# Antioxidant Capacity and Antigenotoxic Effect of *Hibiscus sabdariffa* L. Extracts Obtained with Ultrasound-Assisted Extraction Process

**Gregorio Iván Peredo Pozos [1], Mario Alberto Ruiz-López [2], Juan Francisco Zamora Nátera [2], Carlos Álvarez Moya [3], Lucia Barrientos Ramírez [4], Mónica Reynoso Silva [3], Ramón Rodríguez Macías [2] , Pedro Macedonio García-López [2], Ricardo González Cruz [4], Eduardo Salcedo Pérez [2] and J. Jesús Vargas Radillo [4],***

[1] Doctorate in Sciences in Biosystematics, Ecology and Management of Natural and Agricultural Resources (Bemarena), University of Guadalajara, Nextipac, Zapopan 45510, Mexico; g82ipp@gmail.com

[2] Department of Botany and Zoology, University of Guadalajara, Nextipac, Zapopan 45510, Mexico; mruiz@cucba.udg.mx (M.A.R.-L.); jfzamora@cucba.udg.mx (J.F.Z.N.); ramonrod@cucba.udg.mx (R.R.M.); macedonio.garcia@academicos.udg.mx (P.M.G.-L.); eduardo.salcedo@academicos.udg.mx (E.S.P.)

[3] Department of Molecular and Cellular Biology, University of Guadalajara, Nextipac, Zapopan 45510, Mexico; calvarez@cucba.udg.mx (C.A.M.); monica.reynoso@cucba.udg.mx (M.R.S.)

[4] Department of Wood, Pulp and Paper, Cucei, University of Guadalajara, Nextipac, Zapopan 45510, Mexico; lbarrien@cucei.udg.mx (L.B.R.); ricagcruz@gmail.com (R.G.C.)

* Correspondence: jvargasr@dmcyp.cucei.udg.mx; Tel.: +33-3682-0110

**Abstract:** *Hibiscus sabdariffa* (Roselle) is in high demand worldwide due to its beneficial health properties owing to the polyphenols content, mainly in the flower calyx. The objective of this study was to find the best conditions (time and liquid: solid ratio) to extract polyphenols from Roselle using Ultrasound-Assisted Extraction (UAE) (40 kHz, 180 W), with ethanol how solvent; as well as determine the yield of phenols, anthocyanin, flavonoids, tannins, antioxidant activity (DPPH) and antigenotoxic effect (comet assay). A traditional solid-liquid extraction was applied as a reference. Extraction times of 40 and 60 min resulted in the highest polyphenols (13.019 mg GAE/g dry weight (dw)), flavonoids (4.981 CE/g dw), anthocyanins (1.855 mg Cya3GE/g dw), and tannins (0.745 CE/g dw) recoveries and an antioxidant activity (DPPH) of 74.58%. Extracts from white calyces contained similar amounts of phenols and flavonoids, but very little condensed tannins (0.049 CE/g dw) and practically no anthocyanins. Extracts from red and white calyces, showed antigenotoxic activity and repaired capacity of damage caused by mutagens in human lymphocytes.

**Keywords:** phenols; flavonoids; antioxidant; comet assay; antigenotoxic activity

## 1. Introduction

*Hibiscus sabdariffa* (Roselle, Java, Jute), family *malvaceae*, is an autogamous plant, known in Mexico and Latin America as Jamaican, and in many parts of the world as Roselle or red sorrel [1]. It is an annual or biannual shrub of African origin up to 2 m high, grown in tropical or subtropical areas [2]. Its cultivation is used to obtain fibers and thick and fleshy green, red (the most common) or red dark calyxes [3]. This plant has been used for ancestral times mainly in the preparation of hot and cold beverages with antioxidant properties and that helps in the treatment of chronic diseases [4], for its antihypertensive, antihyperlipidemic, anti-inflammatory, antibacterial and anticancer effect, among others [5]. These effects have been associated to the presence of anthocyanin, flavonoids, phenolic acids and other organic acids specific to *H. sabdariffa* (hibiscus acid) [6]. Roselle, and one of its main

components, delphinidin, has been reported to inhibit melanoma cell growth and metastasis [7]. With around 200 varieties, mainly red and purple, about 97,000 ton of Jamaican flowers are produced [8]. Mexico is the first producer in Latin America, with approximately 19,000 ha of traditional varieties and of white, pink and red creole varieties, with an average yield of 289 kg (dw)/ha [9]. In Mexico, varieties have been developed through natural mutation of the regional creole varieties, including those of white calyces. The characteristic of this variety (name as "white") are green stems, yellow calyces that dry acquire a light brown color, with an average yield of 570 kg dry calyces/ha [9].

On the other hand, combining the method of extraction with the solvent used is important to get efficient extraction of bioactive compounds. The most used methods to extract metabolites from a solid matrix are solid-liquid extractions (S-L) with solvents of different polarity (acetone, methanol, and water among the most common) [10], considering that acetone and alcoholic solutions are more used for phenols extraction. Ethanol is frequently used in extraction of phenolic compounds because it is considered a cheap, non-toxic and efficient green solvent [11]. During extraction step, the plant material swells enhance the accessibility of solvent into plant cells improving extraction and solubilization [11]. However, solid-liquid methods require long extraction times, high energy consumption and use of high solvent volumes [12]. Also, some compounds have an intracellular location, making their extraction difficult, because dissolution and transport processes are hindered [13]. That is why more efficient techniques have been implemented, such as ultrasound-assisted extraction (UAE), among others [14]. UAE can improve the extraction efficiency by promoting the mass transfer due to possible rupture of the cell wall and reduction of particle size by the acoustic effect of cavitation [15]. Through these structural changes and cell disruptions, phenolic compounds can be more easily extracted from the plants [16]. It is a simple, economic alternative, with higher yields, lower energy use and solvent consumption [17].

Since ultrasound-assisted extraction is a convenient method for take-out bioactive compounds from vegetal, the present work was focused on getting phenols from red and white calyces of two *H. sabdariffa* varieties, using ultrasound-assisted extraction at different treatment times, and determine the yield of phenols, anthocyanins, flavonoids, tannins, antioxidant activity (DPPH) and antigenotoxic effect (comet assay). A conventional, traditional solid-liquid extraction was applied as a reference.

## 2. Materials and Methods

### 2.1. Plant Material

Red and white calices of Roselle were obtained from commercial crops of *H. sabdariffa*. The calyces was dried in the environment, ground, sieved and stored in refrigeration for analysis.

### 2.2. Ultrasound-Assisted Extraction (UAE)

An Ultrasonic water bath (510 Branson, 180 W and 40 KHz) was used. Before applying the main extraction stage, a preliminary evaluation was carried out to determine adequate solid: liquid ratio to obtain the highest yield in extracts. For this, two grams of powder dry calyces were placed in a 100 mL flask glass amber with narrow mouth and then different extraction treatments were applied of 20, 30, 40 and 50 mL of 80% ethanol (*v/v*), for 20, 60 and 120 min. UAE was performed at 32 °C. Samples were filtered through filter paper (Whatman 2). The supernatant was dried at 35 °C (Oven, Felisa AR-290D) and weighed, expressing the solids yield in percentage (*w/w*). Samples were prepared and analyzed in triplicate. Product of this study, it was determined that the best ratio was 2 g/40 mL 80% ethanol (solid: liquid ratio 1:20). Since the best solid-liquid ratio was established, the test was carried out to determine the effect of sonication time on the extraction. In total, 2 g of dry and ground calyces of *H. sabdariffa* was loaded into a in a 100 mL flask glass amber with narrow mouth and 40 mL 80% ethanol (*v/v*) was added. UAE was performed at 180 W and 40 KHz, 32 °C for 20, 40, 60 and 120 min. The final extract was obtained by filtering through a Whatman 2 paper and stored at −4 °C for subsequent analysis. Samples were in triplicate.

### 2.3. Solid-Liquid Extraction

In order to compare the performance of UAE extraction, the traditional S-L extraction method was applied. Two dried grams of ground *H. sabdariffa* (red and white) calyces were extracted with 40 mL of 80% (*v/v*) ethanol in a 100 mL flask glass for 72 h, 32 °C, and 180 rpm using an incubator (LabTech Shaking).

### 2.4. Determination of Total Polyphenols

This assay was estimated by Folin-Ciocalteu method [18], with modifications. The extract (1 mL) was mixed with 1 mL of distilled water and 5 mL of Folin Ciocalteu reagent (10% *v/v*). After 8 min it was added 4.0 mL of sodium carbonate (7.5%). The mixture was allowed to stand for 90 min at 25 °C. The absorbance was measured at 765 nm using a UV-Vis spectrophotometer (Velab 5100 UV). The phenolic content was calculated with a standard curve (10–100 µg/mL, $r^2 = 0.991$) for gallic acid, and the results were expressed as milligrams of gallic acid equivalents per gram of dry matter (mg GAE/g dw). Each standard and sample was analyzed in triplicate.

### 2.5. Total Flavonoid Contents

Aluminum chloride assay method [19] was used to determine the total flavonoid content of the samples. An aliquot of 0.5 mL of each extract was mixed with 1.5 mL of methanol, 0.1 mL 10% $AlCl_3$, 0.1 mL of 1 M potassium acetate ($CH_3CO_2K$) (or 1 M sodium nitrite) and 2.8 mL of distilled water. After 30 min at room temperature, the absorbance was measured at 510 nm with a spectrophotomer Uv-Vis (Velab 5100 UV). Total flavonoid content was calculated based on the calibration curve (5–45 µg/mL, $r^2 = 0.994$) and expressed as Catechin equivalents (mg CE/g dry weight). All samples were analyzed in triplicates and the average values were calculated.

### 2.6. Determination of Anthocyanin

Monomeric anthocyanin pigments content was evaluated following the AOAC official method 2005.02 [20]. This pH differential method is based in the change of color of anthocyanin with pH: at pH 1.0 colored oxonium ions are formed, whereas at pH 4.5 predominates the colorless hemiketal form. The difference in the absorbance of the pigments at 520 nm is proportional to the pigment concentration. Briefly, each sample was properly diluted in pH 1.0 buffer (potassium chloride, 0.025 M) and pH 4.5 buffer (sodium acetate, 0.4 M) and absorbance was determined at both 520 and 700 nm. Anthocyanins were calculated as follow:

$$C_A (mg/mL) = \frac{\lceil (A_{510} - A_{700})_{pH1.0} - (A_{510} - A_{700})_{pH4.5} \rceil (MW)(DF)}{(\varepsilon)(l)} \tag{1}$$

where $A$ is the absorbance, MW is the molecular weight of cyanidin-3-glucoside (449.2 g/mol), DF is the dilution factor, $\varepsilon$ = the molar extinction coefficient for cyanidin-3-glucoside (26,900 L/mol/cm), $l$ is standard path length (cm). Anthocyanin concentration was expressed as mg Cyanidin-3-Glucoside Equivalents per gram of dry weight (mg Cya3GE/g dw). Triplicates and the average values were calculated.

### 2.7. Total Condensed Tannin Contents (Proanthocyanidin)

Tannin content was determined by method of Broadhurst et al. (1978) [21] with modification. An aliquot of 50 µL of extract was mixed with 1.5 mL of a solution of vanillin (4% in methanol) and 750 µL of concentrated HCl. After 20 min of incubation at ambient temperature in the dark, the absorbance was read at 500 nm. Catechin was used for the standard reference curve (0–2.5 µg/mL, $r^2 = 0.996$). The condensed tannin was expressed as mg CE/g dw.

### 2.8. DPPH Radical Scavenging Assay

Samples sonicated for different times were analyzed for their free radical quenching activity. The extracts were evaluated by 2,2-Diphenyl-1-picrylhydrazyl (DPPH) [22], with slight modification. In total, 1 mL extract was mixed with a 1 mL of 0.1 mM of DPPH methanol solution. The mixture was kept at room temperature in the dark for 30 min. Control solutions were prepared mixing 1 mL of methanol and 1 mL of 0.06 mM of DPPH in methanol. The absorbance rates were measured at 515 nm and converted into a percentage of antioxidant activity (% Scavenging) in triplicate using the formula:

$$Scavening\% \ = \ \frac{(Abs_{control} - Abs_{sample})}{Abs_{control}} \times 100 \tag{2}$$

DPPH is a stable free radical which mixed with antioxidant (AH) is reduced: $DPPH^* + AH \rightarrow DPPH - H + A^*$ and the change in color from deep violet to pale yellow can be monitoring by the decrease in its absorbance at 515 nm [22]. The reduction of DPPH absorption is indicative of the capacity of the extract to scavenge free radicals. Lower absorbance indicates higher free radical scavenging activity [23].

### 2.9. Antimutagenic Test: Comet Assay

The test was performed according Olive and Banath (2006) [24], with some modifications. Human lymphocytes (0.25 mL) and 2 mL extract of red or white Jamaican calyces was mixed. They were kept on ice for 2 h. Afterward 5 mL phosphate buffer ($NaCl+Na_2HPO_4+NaH_2PO_4+EDTA$) was added. Then sample was centrifuged 10 min at 2500 rpm; the supernatant was removed, and 1 mL of phosphate buffer was applied to the sediment and centrifuging by 10 min. This procedure was repeated with 5 mL of phosphate buffer, taking out the supernatant. Precoated slides (Bioreagent 2-hydroxyethyl agarose, sigma) were coated with 10 μL cell suspension. Slides were kept on ice. A second layer of 0.5% agarose was added and then brought in lysis with 40 μL buffer (2.5 M NaCl; 100 mM EDTA; 10 mM TRIS; 1% (v) Lauryl sarcosinate; 1% (v) Triton-X-100 and 10% (v) DMSO). pH was adjusted to value >9.0 with NaOH. After 6 h of lysis, the buffer was removed and electrophoresis buffer was added (0.3 M NaOH, 1 mM EDTA) for 30 min at 450 mA. Sample was removed from electrophoresis and 80 μL of ethidium bromide (2–7% v) was added, washed with water for 20 min, and stained for 10 min with gelred solution. Then they were analyzed with an Axioskop 40 (Zeiss) epifluorescence microscope with Comet assay system II software (Germany). The length of the comet tail was used as the measure of DNA damage. Ethyl methane sulfonate (EMS) (40 mM) was used as a positive control (mutagen), while a negative control used a preparation that includes only lymphocytes.

### 2.10. Statistical Analysis

A randomized design was used in which sonication time (20, 40, 60, 120 min) was considered as the experimental variable, while the amount of the compounds evaluated (total phenols, anthocyanin, flavonoids, tannins), as the response variable. Measurements were performed at least by duplicate, expressed as mean ± standard deviations ($\overline{x} \mp sd$). Data were analyzed by one-way ANOVA and Fisher's LSD post-hoc test ($p < 0.05$ as statistically significant)

## 3. Results

The study to set the most appropriate solid: liquid ratio, are shown in Figure 1. As seen in the figure, yield of extracts is increased significantly until 1:20 solid: liquid ratio; after which, the yield remains almost constant, appreciating a plateau at 60 and 120 min of treatment. Therefore, this solid-liquid ratio was chosen to measure the effect of sonication time on the extraction, which is discussed later. Regarding this result, it is to be considered that the amount of solvent used is an important factor.

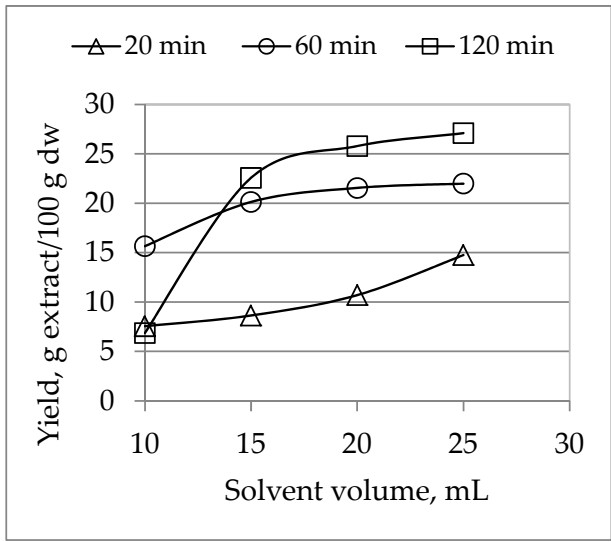

**Figure 1.** Effect of the solvent volume on the yield of extracts.

Once this ratio was established, then the main compounds were extracted at different treatment times, recording the yield of the main compounds and the antioxidant effect by DPPH, as shown in Table 1.

The next are the highest values, all in red Jamaican: Polyphenols 13.019 mg GAE/g dw (60 min sonication); Flavonoids 4.981 mg CE/g dw (sonication time 120 min); Anthocyanins 1.855 mg Cya3GE/g dw (sonication time 40 min), Tannins 0.745 mg CE/g dw (sonication time, 120 min). The results in Table 1 were graphed with the purpose of appreciating and discussing their trend better, as shown in Figure 2.

As can be seen Figure 2, extraction yield of phenols, flavonoids and anthocyanin was increased when the treatment was increased from 20 to 40 min, while at a higher sonication time, the yield is presented a plateau since it remained constant or with slight increases for both types of Jamaica. On the contrary, in the extraction of tannins, 20–40 min treatment was insufficient to obtain the best extraction percentage, so that more sonication time was necessary, which caused a significant increase in yields. Therefore, in red Roselle treatments of 20 and 40, show yields of almost similar tannins, however with 120 min of extraction, the increase was almost 100% compared to that obtained with 40 min.

**Table 1.** Total polyphenols, anthocyanin, tannins and antioxidant activity (DPPH assay) of *H sabdariffa* at different sonication treatment time, solid: liquid ratio 1:20, solvent 80% ethanol, temperature 32 °C, power 180 W, frequency 40 kHz. Data are average of 3 treatments and presented as $\bar{x} \mp s$. Different letters are significantly different ($p < 0.05$) Fisher's Least Significant Difference (LSD), post hoc.

| Time of Sonication | Solids Yield, g/100 g dw | Polyphenols mg GAE/g dw | Flavonoids mg CE/g dw | Anthocyanins mg Cya3G)/g dw | Tannins mg CE/g dw | DPPH, % |
|---|---|---|---|---|---|---|
| | | | Red Calyx | | | |
| 20 | 9.98 ± 0.39 [a] | 11.464 ± 0.90 [a] | 2.201 ± 0.33 [a] | 1.271 ± 0.54 [a] | 0.344 ± 0.16 [a] | 18.40 ± 10.54 [a] |
| 40 | 14.28 ± 2.77 [a] | 12.711 ± 0.11 [b] | 4.186 ± 1.03 [b] | 1.855 ± 1.07 [b] | 0.368 ± 10.15 [a] | 34.36 ± 2.19 [b] |
| 60 | 20.84 ± 2.26 [b] | 13.019 ± 0.12 [b] | 4.416 ± 1.04 [b] | 1.804 ± 1.05 [b] | 0.592 ± 0.02 [b] | 74.58 ± 6.62 [c] |
| 120 | 28.44 ± 2.26 [c] | 12.947 ± 0.18 [b] | 4.981 ± 0.88 [b] | 1.763 ± 1.02 [b] | 0.745 ± 0.038 [b] | 72.07 ± 1.27 [c] |
| (S-L) * | 33.00 ± 1.697 | 65.287 ± 0.015 | 0.448 ± 0.084 | 0.902 ± 0.151 | 0.188 ± 0.002 | 52.89 ± 4.31 |
| | | | White Calyx | | | |
| 20 | 6.64 ± 0.89 [a] | 4.914 ± 0.015 [a] | 0.582 ± 0.48 [a] | 0.00 ± 0.05 [a] | 0.003 ± 0.02 [a] | 12.76 ± 3.35 [a] |
| 40 | 6.42 ± 8.51 [a] | 12.581 ± 0.11 [b] | 3.525 ± 1.34 [a] | 0.003 ± 0.01 [a] | 0.00 ± 0.00 [a] | 8.11 ± 1.60 [a] |
| 60 | 13.74 ± 5.12 [a] | 12.744 ± 0.08 [b,c] | 4.527 ± 0.85 [b] | 0.005 ± 0.00 [a] | 0.021 ± 0.0 [b] | 36.95 ± 0.41 [b] |
| 120 | 21.94 ± 5.80 [a] | 12.953 ± 0.08 [c] | 4.868 ± 2.01 [b] | 0.00 ± 0.07 [a] | 0.049 ± 0.01 [c] | 35.29 ± 5.55 [b] |
| (S-L) * | 20.00 ± 2.828 | 57.244 ± 3.341 | 0.217 ± 0.018 | 0.005 ± 0.00 | 0.000 ± 0.00 | 24.64 ± 12.297 |

* S-L = Solid-Liquid Extraction.

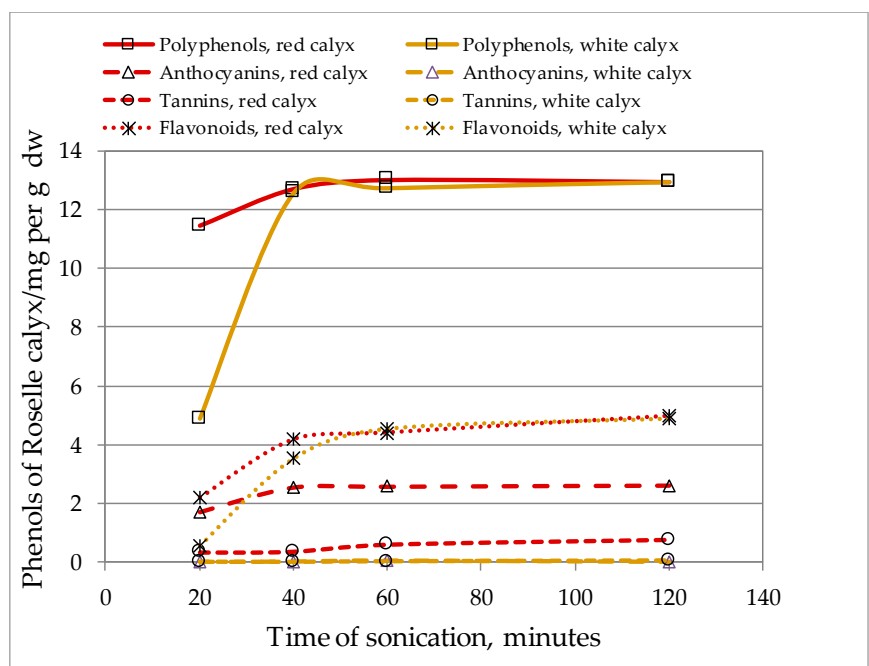

**Figure 2.** Influence of time of sonication over yield of bioactive compounds in the ultrasound-assisted extraction process.

In white Roselle, the tannins were extracted until 60 min of treatment (yield of 0.021 mg CE/g dw), which was increased by 128.8% (0.0492 mg CE/g dw) with 120 min of treatment. Also, red Roselle (red lines) and white Roselle (orange lines) have similar content of polyphenols and flavonoids. However, red Roselle has considerable anthocyanin content and some tannin content, while white Roselle does not show presence of anthocyanin (or it is very low), as well as also a low tannin content.

Antioxidant activity (Table 1) was directly proportional to extract concentration, and therefore, proportional to extraction time. Red *H. sabdariffa* shows twice antioxidant effect that white Roselle. The highest antioxidant activity was with 60 min of sonication, 74.58% for the red variety and 36.95% for white. At longer extraction time (120 min), this property remained constant.

Table 1 shows also the results of traditional S-L extraction. In the S-L extraction up to 5 times more polyphenols were obtained than the best value observed for UAE in red Roselle, and up to 4.4 times higher in white Roselle. Otherwise, UAE extraction yields up to 11.1 times more flavonoids (4.981 mg CE/g dw vs. 0.448 mg CE/g dw, respectively), 2.05 times more anthocyanins and 3.96 times more tannins in red Jamaica, while in white Jamaica 22.43 times more flavonoids and 0.049 times more tannins are observed (0.049 mg CE/g dw vs. 0.00 mg CE/g dw). Likewise, the antioxidant activity of the extracts obtained with UAE was 1.41 times higher in red Jamaica (74.58% vs. 52.89%) and 1.50 times higher in white Jamaica, when compared against antioxidant activity of S-L extracts.

In respect to comet assay, our results were: 9.98 μm by red Roselle; 10.15 μm white Roselle; 15.25 μm positive control; and 9.3 μm negative control.

## 4. Discussion

The central theme of this study was to establish the best conditions for the ultrasound-assisted extraction from *H. sabdariffa* calyces. For this, we first determined the best solid: liquid ratio, and then used this to assess the impact of sonication time. Increasing solvent volume resulted in higher dissolution capacity. In addition, larger solvent volumes facilitate the dissolution of the active compounds and result in significant mass transfer and accelerated diffusion [25]. Also, sonication enhances hydration and fragmentation of the material, facilitating mass transfer without significant solvent decomposition [26]. Some studies used a solid: liquid ratio like the one chosen by us. For instance, in the extraction of *Nephelium lappaceum* L. (rambutan), a fruit common in Southeast Asia, with

ultrasound assisted extraction at 20 W, the highest anthocyanin and polyphenols yield was attained at 50 °C with 50 W power, a 1:18.6 g/mL solid: liquid optimum ratio and 20 min extraction time [27]. Also, the antioxidant activity improved as the liquid: solid ratio increased from 10:1 to 40:1 mL/g, reaching maximum at the 40:1 mL/g ratio and plateauing at higher ratios [28]. As we increase the solvent, the contact between the solids and solvent increases and the diffusion of the antioxidants compounds out of the plant cells is enhanced [29]; however, when the diffusion process reaches its equilibrium, the extraction process remains unchanged [28].

The extraction of compounds from the plant material and, consequently their yield, also depends on their location within the cell wall and their solubility in the solvent used, which would explain why longer extraction times were required for tannins. Ultrasound may facilitate the extraction of active compound located intracellularly by creating wave cavitation bubbles and shorth waves that results in changes in temperature, pressure [30], cell disruption and enhance mass transport [31]. This would also explain the greater and better extraction of the compounds by UAE when respect to S-L extraction, except in the case of total phenols (Table 1). Phenolic compounds are more often associated with proteins, carbohydrates (glycosides), terpenes, chlorophyll, lipids and inorganic compounds [32], which would explain the high percentage yield obtained in alcoholic extracts, due to the ability of these solvents to dissolve endogenous compounds together with phenols [33]. Methods that use long periods (72 h, for example), as frequently occurs in the S-L process, should report a greater value of total phenols than those that use short periods (1.5 h for example) such as UAE. Also, the high chemical diversity of phenols and its association with other compounds, with several thousand molecules having a polyphenol structure, could be the cause of the similar yield of total phenols observed for UAE extraction, as well as the little difference (higher in red variety) for S-L extraction, in red and white *H. sabdariffa* (Table 1), in spite the absence of anthocyanins and the lower tannin yield in the white variety. Other studies found slightly higher phenolic yield in light red calyxes than in white calyces (15.5 vs. 13.5 mg/g dw), as well as a significantly higher content in dark red calyxes (36.50 mg/g dw) (Table 2). The white varieties of Roselle highlight for their content of phenolic acids (caffeic, protocatechuic, chlorogenic), substantially greater than those shown by the variety of colorful calyces [34].

**Table 2.** Phytochemicals in calyx of Roselle (*H. sabdariffa*) extracted with ultrasound assisted and solid-liquid methods.

| Extraction Conditions | Solid Yield % | Phe | Fla | Anth | Tan | DPPH % | Reference |
|---|---|---|---|---|---|---|---|
| | | mg/g dw[(*)] | | | | | |
| *Ultrasound assisted extraction* | | | | | | | |
| - 500 mg, 50 mL (ratio 1:10), Ethanol-water 80:20, 30 °C, 20 min | — | 6.90 | 4.05 | 17.53 | — | — | [35] |
| - Red dried calyces 1.5 g, solid: liquid ratio 230 g/Lt, ethanol-water (26.1–41.7) % (*v/v*), 426.9 W, 20 kHz, 45 min | — | — | — | 52.94 | — | — | [36] |
| - Roselle dried calyx, solid-liquid ratio 1:25 (g/mL), Temperature 50 °C, time 22.5–45 min, ultrasonic power 80 W | — | — | — | 5.66 | — | — | [37] |
| *Solid- Liquid Extraction* | | | | | | | |
| - 20 g, 250 mL ethanol, 72 h. | 12.8 | 27.6 | 33.8 | — | — | 69.00 | [33] |
| - 2.5 Red Purple, dry calyces, 25 mL Ethanol: water solutions (50:50 and 70:30 %, *v/v*), 2 h. | — | 26.49 | — | 2.21 | — | — | [38] |
| - 2.5 g light red dry calyxes, 100 mL water, boiled 15 min | — | 15.5 | — | 4.52 | 0.43 | 65.00 | [34] |
| - 2.5 g dark red dry calyxes, 100 mL water, boiled 15 min | — | 36.50 | — | 10.99 | 1.04 | 85.00 | [34] |
| - 2.5 g white dry calyxes ("Alma blanca"), 100 mL water, boiled 15 min | — | 13.5 | — | 0.20 | 0.24 | 65.00 | [34] |

Phe=Phenols; Fla=Flavonoids; Anth=Anthocyanin; Tan=Tannins; (*) Phenols (mg GAE/g dw); Flavonoids (mg QE/g dw); Anthocyanin (mg Cya3GE/g dw); Tannins (mg QUE/g dw).

Table 2 shows multiple reports of works on *H. sabdariffa*. The concentration of the polyphenols (13.02 mg GAE/g dw) and flavonoids (4.981 mg CE/g dw) of the red variety determined in this study are higher than the 6.90 mg GAE/g dw and 4.05 mg QUE/g obtained with 20 min of sonication [35], and lower than 27.6 mg GAE/g polyphenols and 33.8 mg QUE/g flavonoids obtained with ethanol in a S-L process [33]. Anthocyanin of this work of the red variety (1.855 mg Cya3GE)/g dw) are lower than most of the data in Table 2, except the 2.21 mg Cya3GE/g dw obtained from red purple dry calyces [38]. Also, tannins of this work for red variety (0.745 mg CE/g dw) are higher than the 0.43 QUE/g dw extracted from light red variety [34], but lower than the 1.04 mg QUE/g dw from dark red calyces [34] Regarding white calyces, a slightly lower yield was obtained in phenols (12.953 GAE/g dw), and lower in anthocyanins (0.005 mg Cya3GE/g dw) and tannins (0.049 mg CE/g dw), compared to those obtained from white variety calices with boiling water of 13.5 mg GAE/g, 0.20 mg Cya3GE/g and 0.24 mg QUE/g, respectively [34].

Respect to free radical scavenging activity by DPPH, similar values were obtained to those found by other researchers: 69% for red calyx [33] ; 25–85% for dark red dry calyces and 20–65% for light red white calyx, aqueous extracts, concentrations of 50–250 µg/mL [34]; 5.60, 10.24, 17.44 and 31.28% with polyphenol content of 669.48 and 5012.54 mg GAE/100 g at concentrations of 625, 1250, 2500 and 5000 µg/mL, respectively [39]. In addition, scavenging activities of 2, 11, 23, 53 and 86% for an extract with phenolic content of 41.07 mg GAE/g at concentration of 5, 10, 50, 125 and 250 µg/mL, respectively [40].

The antioxidant activity of extracts depends on the amount and type of polyphenols it contains, with positive correlation between phenolic content and antioxidant activity [41]. Many and diverse compounds have been identified in Roselle with biological properties. It has been noted that flavonoids are the main antioxidant group [42]. Quercetin, myricetin and proanthocyanidins, of the epigallocatechin and epigallocatechin-gallate type catechin, are flavonoids found in *H. sabdariffa* [43]. Anthocyanin (delphinidin 3-O-sambubioside and cyanidin 3-O-sambubioside) are one of the flavonoids with the greatest presence and activity [44], giving several interesting properties in *H. sabdariffa*: An increase in anthocyanin content causes an increase in activity against free radicals; this metabolite contributes 51% of the Roselle antioxidant capacity [45]; they may potentially prevent diseases; give the color of the calyces, dark calyces have more anthocyanin than light calyces, but whites or greens lack these flavonoids [46], reason why this compound have been studied extensively [47]. Although white Roselle calyces had no anthocyanin and tannin content, the antioxidant activity of its extract could be due to the presence of phenolic acids and other polyphenols [34]. Also, tannins are valuable compounds with antioxidant properties, of which they have been identified in aqueous extracts of *H. sabdariffa*, chlorogenic acid [48], gallic acid [49], and the Hibiscus protocatechuic acid [40]. Moreover, it was reported that the extract of Roselle was found to be very high in ascorbic acid content or ascorbate, which is a well-known natural antioxidant [50], and caffeic acid [43].

The prevention of genotoxic damage by *H. sabdariffa* extracts was evaluated using the comet assay. This is a sensitive and rapid method for detecting DNA strand breakdown in individual cells, and have different applications, including human biomonitoring [51]. In this method DNA lysis and electrophoresis are performed using acridine orange for staining the DNA, resulting in an image that looks like a "comet" with a distinct head consisting of intact DNA, and tail, which contains damaged or broken pieces of DNA [51]. Our results showed that *H. sabdariffa* extracts, tested at polyphenol concentration of 13.019 mg GAE/g dw for the red, and 12.953 mg GAE/g dw for white varieties, reduced DNA damage and reversed damage at the cellular level, as evidenced by the tail lengths of 9.98 µm and 10.15 µm, respectively. These lengths were shorter than the 15.25 µm of the sample with mutagen EMS (positive control), but close to the 9.3 µm of the sample without mutagen (negative control). Also, Hibiscus anthocyanins quenched DPPH free radicals, reduced the cytotoxicity induced by tertbutyl hydroperoxide [52] and to attenuate hepatotoxicity in rats as well as to protect DNA damage [53], and cytotoxicity [54].

Finally, the effects of acoustic ultrasound have been studied for many years. It is well accepted that acoustic cavitation result in the creation, expansion, and sudden implosive collapse of bubbles, sometimes referred to as hot spots [55,56], with shock waves of several hundred atmospheres and around 5000 °K [57]. Near of the surface of the plant particles the collapse generates a high-speed jet of liquid [55]. Many of the bioactive compounds are located within the cell wall of vegetables, so that the efficiency of the UAE methods in their extraction would be due to the physical impacts over cell wall. In Figure 3 we present a scheme on the effect of ultrasonic cavitation on the cell wall and the extraction of metabolites. The highly recalcitrant structure (Figure 3a) of lignocellulose (cellulose, lignin and hemicellulose) creates limitations for penetration of solvents [58]. The acoustic cavitation causes disruption and damage of the cell walls (Figure 3b) and provides continuous circulation of solvent into the plant particles, continuous solvent mixing and sometimes particle size reduction, and in addition many hollows are opening [59,60]. The disruption of the cell wall and the penetration of the solvent enhances the extraction and solubilization of bioactive compounds (non-structural components).

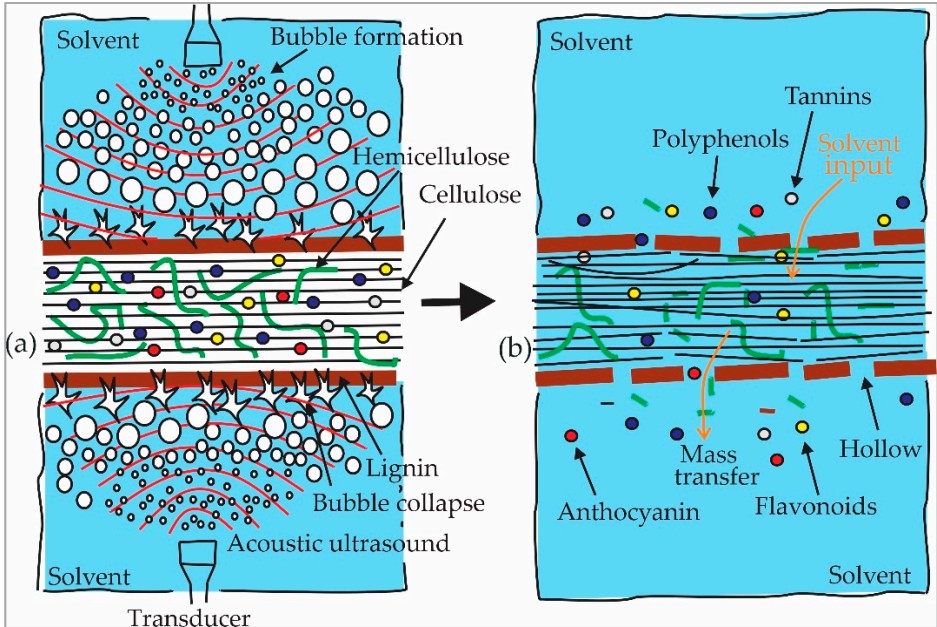

**Figure 3.** Effect of acoustic cavitation on plant cell wall during the extraction of bioactive compounds by ultrasonic assisted extraction method. (**a**) Cell wall during treatment, (**b**) cell wall after UAE process.

UAE is an advanced method of compound extraction, contemplate as a green and sustainable process. It is a clean method that allows high yields with low solvent consumption (generally safe (GRAS) solvents), and short extraction times, therefore lower energy consumption than conventional methods. Laboratory scale calculations establish that the method of extraction by UAE would be the process with lower energy consumption and lower $CO_2$ emission compared to extraction by maceration or Soxhlet, so it can be considered an "environmentally friendly" method, suitable to laboratory level, but scale-up proper for sustainable industrial application [60]. These methods are considered the most feasible and economically profitable in the large scale, due to the minimum detrimental effect on the extracted compounds, they are easy to install, with competitive energy costs that require low maintenance. However, it is important to optimize the variables implied in the process at laboratory scale, because the process is sensitive [61]. The main companies that have developed and commercialized large-scale ultrasound devices are Hielscher (Germany), offering ultrasonic probes with power range of 500 to 16,000 Watts; and REUS (France), that offer ultrasonic baths of 500 to 1000 L [60]. UAE represents an alternative for the extraction of phenolic compounds at industrial level, as it has been successfully applied in several large-scale experiments [62].

## 5. Conclusions

Ultrasound-assisted extraction allowed extracts from red and white *H. sabdariffa* varieties with yield, antioxidant and antigenotoxic activity within average values, with the advantage of using a common and accessible Ultrasonic device (water bath), ethanol as solvent and relatively short times of extraction. Except for tannins, the best yields were obtained with solid: liquid ratios ≥ 1:20 and 40 to 60 min extraction times. Extracts from both varieties prevented DNA damage, in an In vitro test, and had antioxidant activity. UAE methods also recorded better yields of biocompound relative to the traditional S-L extraction process, despite using significantly less time lapses. Extracts from the white Jamaica variety, although lacking anthocyanin, had antioxidant activity—possibly due to the high content of phenols and flavonoids.

**Author Contributions:** G.I.P.P. carried out the experimental works and manuscript preparation; M.A.R.-L., J.F.Z.N. and P.M.G.-L. coordinated and reviewed the status of the extraction and quantification of bioactive compounds; L.B.R. directed and evaluated the DPPH antioxidant activity tests; C.A.M. and M.R.S. directed and evaluated the antigenotoxic comet assay; J.J.V.R. participate in the experimental phases and in the preparation and revision of the manuscript; R.G.C. revised of figures and tables; R.R.M. and E.S.P., literature and manuscript review. The authors approve the final manuscript. All authors have read and agreed to the published version of the manuscript.

**Funding:** This study received the support of the National Council of Science and Technology, Conacyt, and the Ministry of Public Education (SEP), through a scholarship for doctoral studies in Biosystematic, Ecology, and Management of Natural and Agricultural Resources (Bemarena).

**Conflicts of Interest:** The authors declare no conflict of interest.

## Abbreviations

| | |
|---|---|
| UAE | Ultrasound-assisted extraction |
| dw | Dry weight |
| DPPH | 2,2-Diphenyl-1-picrylhydrazyl |
| CE | Catechin Equivalents |
| Cya3GE | Cyanidin-3-Glucoside equivalents |
| QUE | Quercetin Equivalents |
| μg | microgram |
| EMS | Ethyl methane sulfonate |
| DMSO | dimethylsulfoxide |
| S-L | Solid-liquid extraction |
| CA | Concentration of Anthocyanin |
| EDTA | Ethylenediaminetetraacetic acid |
| TRIS | trisaminomethane |

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
