# Peer review of "Antioxidant Capacity and Antigenotoxic Effect of Hibiscus sabdariffa L. Extracts Obtained with Ultrasound-Assisted Extraction Process"

_applsci, doi:10.3390/app10020560_

Round 1

Reviewer 1 Report

This paper investigated the influence of ultrasound-assisted extraction (UAE) on the rapid and selective recovery of the phenolic compounds from red and white calyces of Hibiscus sabdariffa (Roselle). As the best extraction solvent, 80% of ethanol was obtained. Further extractions of red and white calyces of Rosella were carried out to obtain the best liquid: solid ratio (10, 15, 20 and 25) and time of sonication (20, 40, 60 and 120 min) under UAE at 180 W and 40 kHz, 32 oC. The effectiveness of UAE was tested by the determination of the yield of extracts (as g/100 g DW), total polyphenols (as mg GAE/g DW), total flavonoids (as CAE/g DW), total anthocyanins (as mg Cya3G/g DW), total tannins (as mg CA/g DW) and antioxidant capacity (as % DPPH scavenging).

Finally, extracts with a polyphenol concentration of 13.019 mg GAE/g DW in Rosella red and 12.953 mg GAE/g DW in white variety were tested against the prevention of genotoxic damage by the Comet assay method.

The article is of interest to the scientific community because is designing a green and efficient extraction process of phenolic compounds from H. sabdariffa as valid alternatives to traditional methods.

However, this work has numerous flaws that need to be addressed before publication. I believe that this work will be better with suggested revisions.

Abstract

Pg 1, L20-24: in the sentence “The objective…”, replace “(time and ratio material/liquid)” into “(time and liquid: solid ratio)”. In addition, this sentence should be reformatted.

Material and Methods

Pg 2, L29: in the sentence “Extracts from… ” should be added which of the extracts from red and white calyxes showed antigenotoxic activity".

This subsection will describe the chemicals and assays used in the work.

Pg 2, L84: the liquid: solid ratio instead solvent volume (20, 30, 40 and 50 mL) is more appropriate to use.  

Pg 2, L85: In the sentence, “Samples were filtered…”, specify which filter you used.

Pg 2, L85: In the sentence, “The supernatant was dried…” specify which equipment was used for “drying”.

Pg 3, in the subsection 2.5 Determination of Anthocyanin, the equation should be numbered.

Pg 4, L141: the equation should be numbered. The description of the equation is needed.

Pg 4, L142-146 should be omitted. The method description is not required.

Pg 4, L167: the title of subsection should be corrected into "2.9. Statistical analysis".

Results

Figure 1: should be corrected as liquid: solid ratio vs. yield. The description of Figure 1 should be adapted according to the corrected Figure 1.

Table 1: the description of the table should be corrected and complemented (with 80% ethanol).

Figure 2: should be corrected (Anthocianins into Anthocyanins). In addition, Table 1 and Figure 2 present the same data and the authors should decide only one of them.

Discussion

The Discussion should be re-written taking into account a comparison of data and extraction conditions for polyphenols, flavonoids, anthocyanins, tannins, and antioxidant activity. The discussion should be clearly written. Why Refs. 34, 36, 38 and 39 are not listed in Table 2? In addition, I do not find the value of flavonoids (4.657 mg CA/g DW) in Table 1 as mentioned in the Discussion (Pg 7, L251).

Additionally:

The results of the genotoxicity test were not presented; however, two obtained results (length tail 9.98 µm for red Roselle and 10.15 µm for white Roselle) were discussed.

The Dry weight (DW) mark should be uniform in the text.

References should be described according to Instructions for Authors.

Many typescript errors should be corrected in the text.

The Graphical abstract is missing.

Conclusion:

This manuscript can be accepted after the implementation of suggested changes and corrections.

Author Response

Appreciable Dr. We apply the corrections indicated by you to the document, which are detailed point-by-point in the attached file (cover letter). We apply a review to detect and correct possible errors. We appreciate your opinion on the manuscript and its very precise observations. Greeting and our recognition and thanks for the review of the article.

Reviewer 2 Report

You should provide a list of abreviations The extract from raw material was made with EtOH 80%, it is non representative of what could be the traditionnal use of Hibiscus tea (a water extract would be more representative) Characterization of the extract can be improved :Folin method is widely used for the total polyphenols content, nevertheless it is a method giving an indication of the reducing ability of the extract, not its polyphenol content (a high amount in Vitamin C can lead to false results) ; vanilin method is not the best one for the tanin content, the bate Smith test is more accurate. A chromatographic method (HPLC/MS) should be used to characterize/quantify the main molecules present in the extract.  Despite a lower content in anthocyanins and tanins of the white calyx, and a similar content in other flavonoids, the total polyphenol content is identical in white and red calix (Fig.2), which is non logical. Antiox activity (DPPH) is only in vitro, and non representative of what could be the in vivo effect, because of the low bioavailability of a lot of polyphenols : other mechanisms of actions (beside the antiox effect) should be considered to explain the the beneficial effects of the Hibiscus tea. 

Author Response

Dear Dr. We read your comments carefully. Basically, we believe that the techniques applied in our study were adequate, in the cover letter you will find our argument in this regard; In addition, it is not possible implement the complementary analyzes to those presented in our work and for you suggested, due to the dates set (8 days) to respond to your comments and reload the corrected document. However, we include new text in the manuscript to satisfy some of its observations and that you will find detailed in the attached file (cover letter). We believe that the corrected manuscript contains information of potential interest on ultrasound-assisted extraction, a central theme of this study and proposed for the special issue "Ultrasound in Extraction Processing", and also for H. sabdariffa. Greetings and our thanks for the review of the article. For us it would be a great satisfaction communicate our results through Applied Sciences.

Reviewer 3 Report

The topic is interesting but the main settings of experiment are poor. Namely, to consider the UAE as the promissing extraction technique and to point out it advantages in terms of better extraction yield of analysed bioactive compounds in shorter time period some control must be done, for example conventional (classical) solid-liquid extraction technique could be used as control sample to compare the efficiency of UAE. Also, in study were considered just two parameters characteristic for UAE: the ratio of sample material and solvent and time of extraction, but for better understanding and results of the experiment other UAE parameters are very important to consider, for example the solvent type, besides only 80 % EtOH (v/v) other alcohol (EtOH) concentrations could be considered (50, 30 %), or even another solvent type (as distilled water characterized as GRAS safe). 

Considering the all above mentioned, I recommend authors to upgarde the experiment and reconsider to reapply it for publication.  

Author Response

Appreciable Dr., due to the time (8 days) to correct and reload the document, it is not possible to reformulate the experimentation including more variables, which would require months of work. In the attached file (cover letter) you will find the argument in our favor about the type and number of the experimental variables used in our work. However, if it was possible to include in the manuscript, as a control, a classic extraction by maceration (solid-liquid) with conditions similar to those used in UAE, and discuss it. We believe that the study can contribute with potentially interesting results and information to the subject of the special issue “Ultrasound in Extraction Processing”. Receive a cordial greeting and our thanks for the revision of the manuscript.

Round 2

Reviewer 1 Report

The authors have made most of the indicated changes so the quality of the article has improved considerably. However, I still have some comments as follows:

Pg 1, L25: in the sentence “Extraction times…(13.019 mgGAE/g…), replace with “Extraction times…(13.019 mg GAE/g…)”.

Pg 1, L30: replace the word “repared” with “repaired”.

In the whole text replace “…° C” with “…°C”.

Pg 4, L146: replace the part “(mg Cya3GE)/gdw)”, with “(mg Cya3GE)/g dw)”.

The description of Figure 1 must be uniform with Figure (especially x-axis).

Pg 6, L222: transfer the part of the sentence "In white Roselle ..." from Pg 6 to Pg 8, where is the rest of the sentence (in L243).

In Figure 2, the title of the y-axis "mg compound/g (Roselle calyx) dw" should be replaced with "Phenols of Roselle calyx/ mg per g dw"

Pg 8, L252: put space after “Table 2…”

Pg 8, L255: replace the part of the sentence "(4,981 mg CE/g dw..." with "(4.981 mg CE/g dw..."

Pg 9, L295: “...(table 2)…” must be replaced with “…(Table 2)…”

Pg 9, L301: in the sentence “In order…” put space after Table 2.

Pg 9, L303: replace “…(4.981mg CE/g dw)…” with “…(4.981 mg CE/g dw)…”

Pg 9, L304: replace “…dw and 4.05 mg QUE/g…, and Lower than …” with “…dw and 4.05 mg QUE/g…, and lower than …”

Pg 9, L306: the capital letter in the word "Table" must be uniform in the whole text.

Pg 10: transfer the title of Table 2 above Table 2.

“in vitro” (L387) and "in vivo" must be italic in the whole text (L432, L562)

“vs” should be italic in the whole text (L255, L257, L258, L297).

Author Response

Response

Pg 1, L25: in the sentence “Extraction times…(13.019 mgGAE/g…), replace with “Extraction times…(13.019 mg GAE/g…)”.

Reply. The replacement was made: Word file (L26); Pdf file (L25)

Pg 1, L30: replace the word “repared” with “repaired”.

Reply. The replacement was made: Word file (L36); Pdf file (L30)

In the whole text replace “…° C” with “…°C”.

Reply. The correction was applied: Word file (L101, L109, first paragraph table 2); Pdf file (L101, L113, first paragraph table 2).

Pg 4, L146: replace the part “(mg Cya3GE)/gdw)”, with “(mg Cya3GE)/g dw)”.

Reply. The replacement was made: Word file (L151); Pdf file (L146)

The description of Figure 1 must be uniform with Figure (especially x-axis).

Reply. It was changed in the description of the figure 1 “solid: liquid ratio” by “solvent volume”: Word file (L212); Pdf File (L206)

Pg 6, L222: transfer the part of the sentence "In white Roselle ..." from Pg 6 to Pg 8, where is the rest of the sentence (in L243).

Reply. The part of sentence "In white Roselle ..." was changed as indicated: Word file (L228 to L252); Pdf File (L222 to Line 243)

In Figure 2, the title of the y-axis "mg compound/g (Roselle calyx) dw" should be replaced with "Phenols of Roselle calyx/ mg per g dw"

Reply. This recommendation is accepted, so the title of the y-axis of the figure 2 were replaced as indicated

Pg 8, L252: put space after “Table 2…”

Reply. The text was revised, and it was found that the space after Table 1 was already applied. Word file (L228 to L252); Pdf File (L262)

Pg 8, L255: replace the part of the sentence "(4,981 mg CE/g dw..." with "(4.981 mg CE/g dw..."

Reply. The indicated correction was applied: Word file (L265); Pdf File (L255)

Pg 9, L295: “...(table 2)…” must be replaced with “…(Table 2)…”

Reply. The indicated correction was applied: Word file (L314); Pdf File (L295)

Pg 9, L301: in the sentence “In order…” put space after Table 2.

Reply. The text was revised, and it was found that the space after Table 2 was already applied. Word file (L327); Pdf File (L301)

Pg 9, L303: replace “…(4.981mg CE/g dw)…” with “…(4.981 mg CE/g dw)…”

Reply. The replacement was made as indicated: Word file (L329); Pdf File (L303)

Pg 9, L304: replace “…dw and 4.05 mg QUE/g…, and Lower than …” with “…dw and 4.05 mg QUE/g…, and lower than …”

Reply. The replacement was made as indicated: Word file (L331); Pdf File (L304)

Pg 9, L306: the capital letter in the word "Table" must be uniform in the whole text.

Reply. The manuscript was revised and the word “table” was uniformed, applying the capital letter: Word file (L258, L314, L336); Pdf File (L248, L295, L306)

Pg 10: transfer the title of Table 2 above Table 2.

Reply. The title of Table 2 was transferred to the top of the same: Word file (L276); Pdf File (L373)

“in vitro” (L387) and "in vivo" must be italic in the whole text (L432, L562)

Reply. The manuscript was revised and homogenized "in vivo" and “in vitro” by writing it in italic form: Word file (L433, L483 (Ref 7), L631 and L632 (Ref 48)); Pdf File (L387, L432 (Ref 7), L 562 and L563 (Ref 48))

“vs” should be italic in the whole text (L255, L257, L258, L297).

Reply. “vs” was changed to italian form “vs.”: Word file (L265, L267, L268, L316); Pdf File (L255, L257, L258, L297)

Dear Dr., thank you very much for your comments and observations because they are very helpful and allow us to improve our article

Reviewer 2 Report

RAS

Author Response

Dear Dr., we revise the manuscript again to avoid possible errors. Your comments and suggestions during this process have been very adequate and have helped improve its quality. We appreciate the opportunity that is being provided. Thank you

Reviewer 3 Report

In Conclusion sector please correct the state that UAE is " a cheap..." as advantage of UAE, because in overall it is not a cheap method especially at the industry scale. I understand that for other UAE variables there is enough time so it is ok just to include a control sample (solid-liquid extraction).

Author Response

In Conclusion sector please correct the state that UAE is " a cheap..." as advantage of UAE, because in overall it is not a cheap method especially at the industry scale. I understand that for other UAE variables there is enough time so it is ok just to include a control sample (solid-liquid extraction).

Reply. We agree with your suggestion, so the word "cheap" of conclusions was eliminated, since indeed, the process is not cheap at the industrial level. Instead the words "accessible" and "common" were included in reference to the ultrasonic bath.

Dear Dr., thank you very much for accepting our responses to the observations of the first round, particularly with regard to the variables considered in our work. Also thank you for your comments regarding the incorporation of the traditional extraction process, as a method of comparison, which allowed us to improve our article.